# Biology and Prevalence in Northern Italy of *Verrallia aucta* (Diptera, Pipunculidae), a Parasitoid of *Philaenus spumarius* (Hemiptera, Aphrophoridae), the Main Vector of *Xylella fastidiosa* in Europe

**DOI:** 10.3390/insects11090607

**Published:** 2020-09-07

**Authors:** Giulia Molinatto, Stefano Demichelis, Nicola Bodino, Massimo Giorgini, Nicola Mori, Domenico Bosco

**Affiliations:** 1Dipartimento di Scienze Agrarie, Forestali e Alimentari, Università degli Studi di Torino, Largo Paolo Braccini, 2, 10095 Grugliasco, Italy; giulia.molinatto@unito.it (G.M.); stefano.demichelis@unito.it (S.D.); 2CNR—Istituto per la Protezione Sostenibile delle Piante, Strada delle Cacce, 73, 10135 Torino (TO), Italy; nicola.bodino@ipsp.cnr.it; 3CNR—Istituto per la Protezione Sostenibile delle Piante, Piazzale Enrico Fermi, 1, 80055 Portici (NA), Italy; massimo.giorgini@ipsp.cnr.it; 4Dipartimento di Biotecnologie, Università di Verona, Via della Pieve 70, 37020 San Floriano (VR), Italy; nicola.mori@univr.it

**Keywords:** spittlebug, pipunculid fly, big-headed flies, *Neophilaenus campestris*, *ITS2*, *COI*

## Abstract

**Simple Summary:**

The meadow spittlebug, *Philaenus spumarius* is the major vector of *Xylella fastidiosa* in Europe. The spread of *X. fastidiosa* depends almost exclusively on insect transmission, and therefore, it is vital to keep vector populations low. To achieve this goal, natural enemies should be identified and their efficacy evaluated. The aim of this work was to assess the presence and abundance of a parasitoid fly, *Verrallia aucta*, in field-collected spittlebugs. At first, we developed a new species-specific molecular tool (PCR) to identify the parasitoid, then we estimated the parasitization rate in different sites of northern Italy using both PCR and the dissection of insect bodies. Finally, we established a small-scale rearing to describe the life cycle of the fly on its spittlebug host. *Verrallia aucta* is quite common in northern Italy but displayed low prevalence, reaching a maximum parasitization rate of 17.5% in vineyards of the Piemonte region. The fly has one generation per year, lays eggs in newly emerged adults of spittlebugs, and develops inside the host throughout the summer. The mature larva abandons the dead victim at the beginning of autumn and pupates in the soil where it overwinters.

**Abstract:**

The meadow spittlebug, *Philaenus spumarius* (L.) (Hemiptera Aphrophoridae), the main vector of *Xylella fastidiosa* Wells et al. in Europe, has few known natural enemies. The endoparasitoid *Verrallia aucta* (Fallén) (Diptera, Pipunculidae) was first noticed a long time ago but very little is known about its biology and prevalence. In this study, the presence and prevalence of *V. aucta* were investigated in different regions of northern Italy, both in plain–foothill and montane zones. Parasitic larvae were identified by the dissection of spittlebug adults, *P. spumarius* and *Neophilaenus campestris* (Fallén), and by a new species-specific molecular tool targeting the *ITS2* and *COI* genomic regions, developed in this work. A small-scale rearing was set up to gain information on the life cycle of *V. aucta* on its main host *P. spumarius*. During the four-year investigation (2016–2019) the pipunculid parasitoid displayed low prevalence, reaching a maximum parasitization rate of 17.5% (calculated over the adult spittlebug season) in vineyards of the Piemonte region. Over the whole period, no significant difference in the prevalence was found between male and female spittlebugs. Collected data and rearing observations suggest that *V. aucta* is monovoltine and synchronous with *P. spumarius*, laying eggs in newly emerged adults, developing as an endoparasitoid through two larval stages during the whole summer, and overwintering as a pupa in the soil.

## 1. Introduction

Xylem fluid-feeding insects (order Hemiptera, sub-order Auchenorrhyncha) belong to three superfamilies: Cercopoidea (spittlebugs or froghoppers), Cicadoidea (cicadas), and Membracoidea (which includes a single xylem fluid-feeding subfamily, the Cicadellinae, known as sharpshooters) [1]. These insects, namely spittlebugs and sharpshooters, are the vectors of *Xylella fastidiosa* Wells et al. to a high number of plant species. Differently from those in America, in Europe, where *X. fastidiosa* is considered of recent introduction, all identified vectors are spittlebugs in the family Aphrophoridae [2,3]. Indeed, their importance has greatly increased since the discovery of the bacterium, as in Europe these insects are not considered direct pests, unless present at very high population levels [4,5]. *Philaenus spumarius* (L.), the meadow spittlebug, is by far the most important vector species, as shown by its major role in the devastating epidemic of Olive Quick Decline Syndrome (OQDS), caused by *X. fastidiosa* subsp. *Pauca* ST53, in the Apulia region of Italy [3,6], and in the spread of Pierce’s disease in vineyards of Mallorca, Spain [7]. Furthermore, *P. spumarius* is the dominant species among spittlebugs, ubiquitous and locally very abundant [8] and although all xylem sap feeders should be regarded as potential vectors [9], this species is considered to play a key role responsible for the spread of *X. fastidiosa* under cropping and non-cropping conditions in Europe [6,7,10].

While the amount of information on the phenology and ecology of *P. spumarius* and closely related species is rapidly increasing [11,12,13,14,15,16,17,18,19], the literature on biological control agents of these insects is very scarce. Indeed, it is restricted to reports on the presence of egg parasitoids of the genus *Ooctonus* (Hymenoptera Mymaridae) and *Centrodora* (Hymenoptera Aphelinidae), the endoparasitoid *Verrallia aucta* Fallén (Diptera Pipunculidae) and few uncharacterized entomopathogenic fungi (revised in [8,14]). Interestingly, very recently, one of the oophagous parasitoids recorded from US, *Ooctonus vulgatus* Haliday, has been found commonly and locally abundant in Corsica island of France [20]. Finally, the newly introduced predatory bug *Zelus renardii* Kolenati (Hemiptera, Reduviidae) has been proposed for the inundative control of *P. spumarius* with the aim of suppressing the spread of *X. fastidiosa* in olive groves of Apulia [21]. However, the mass release of such a generalist predator may be risky for local biodiversity, especially for beneficial arthropods [22], while its efficacy under field conditions has yet to be demonstrated. Among generalist predators, spiders have been considered as potential valuable biocontrol agents of *P. spumarius.* A protocol aiming at facilitating the selection of spider species that could represent potential natural enemies of *P. spumarius* in olive crops has been recently proposed [23].

As for the biology of the pipunculid parasitoid, Whittaker [24] stated that *V. aucta* develops only in spittlebug adults parasitized soon after the emergence from the spittle cocoon. Furthermore, the parasitoid larval stages, likely two, kill the host at the end of its life cycle. Thus, not preventing the transmission of *X. fastidiosa* by infectious adults, it still has a potentially important suppressive effect on the population, as parasitized adults are sterile [25]. This species, according to the Fauna Europaea database [26], appears to be quite widespread but very poorly studied. We know from Whittaker [25] that *V. aucta* parasitize both *P. spumarius* and *Neophilaenus lineatus* (L.), and that parasitism rates can be relatively high in England: on average 31% in females and 46% in males over four years.

The aim of this study was to gain information on *V. aucta* presence and prevalence in the *P. spumarius* and *Neophilaenus campestris* (Fallén) populations of northern Italy and on the parasitoid life cycle in relation to the spittlebug host. Many spittlebug adults from different sites of northern Italy were examined for the presence of *V. aucta* either by dissection or by a new PCR assay designed for the specific detection of the parasitoid in the spittlebug body. The high sensitivity of this latter method allows for the identification of the parasitoid even in the very early stages of parasitization, when the larva is very small or only the egg is present, and allows the detection of the parasitoid in multiple samples of grouped host specimens at the same time, thus avoiding dissecting many insects individually. Together with available information on other parasitoids and predators, this work contributes to the description of the community of natural enemies that should be preserved and enhanced in order to limit spittlebug populations.

## 2. Materials and Methods

### 2.1. Insect Collection and Dissections

#### 2.1.1. Field Collection

Adults of *P. spumarius* were collected with a sweep net in the years 2016–2019 in different localities of Piemonte, Liguria, Veneto and Trentino-Alto Adige regions of Italy (Table 1). Sampling took place in the Piemonte and Liguria regions in all four years, while in Trentino-Alto Adige and Veneto regions’ samples were collected in 2018 and 2019, respectively. The different collection sites were grouped by geographic proximity (a letter code was assigned to each group) and discriminated according to their elevation (low, plain–foothill and high, montane) (Table 1). Sites in the montane zones were located between 1000 and 2200 m above sea level (a.s.l.) in the Alps, while those in the plain–foothill zones were located below 350 m a.s.l.

Adults of *N. campestris* were collected with a sweep net in 2018 and 2019 in six localities of the Piemonte region, both in the plain–foothill (Chieri and Grugliasco, Table 1, Site code: C; Asti and Cocconato, D; Paderna, F) and in the montane zone (Prali, G).

#### 2.1.2. Dissections

Adults of *P. spumarius* were sampled for dissection over all the adult season (beginning of May until December) in the years 2016–2019. *Neophilaenus campestris* adults were collected from the end of May to September in 2019. Overall, 492 *P. spumarius* and 26 *N. campestris* were dissected for the presence of *V. aucta* parasitoids. Dissection was performed in 1% physiological solution.

### 2.2. Molecular Detection of Verrallia aucta

Samples analyzed by PCR for the presence of the parasitoid were collected from June to September 2018 and 2019. Overall, 922 adults of *P. spumarius* and 37 adults of *N. campestris* were analyzed by PCR.

#### 2.2.1. Primer Design

Nucleotide sequences of *V. aucta* mitochondrial partial cytochrome oxidase subunit 1 (*COI*) gene (accession: FM178086.1) and nuclear partial non-coding internal transcribed spacer 2 (*ITS2*, accession: FM178155.1) were retrieved from the NCBI GenBank [27]. These regions were chosen because they are species-specific within *Verrallia* species according to Kehlmaier and Assmann [28]. Sequences were submitted to Primer3 software [29] and one primer pair was selected for each target (Table 2). Primers were tested in conventional PCR on DNA extracted from the larvae of *V. aucta* isolated from parasitized adults of *P. spumarius* and *N. campestris,* and in order to verify their specificity, on DNA extracted from non-parasitized adults of *P. spumarius*. PCR products were purified with the DNA Clean & Concentrator-25 kit (Zymo Research, Irvine, CA, USA) following the manufacturer’s instructions and send to BMR Genomics (Padova, Italy) for sequencing.

#### 2.2.2. DNA Extraction

Male and female adults of *P. spumarius* were separated under a stereomicroscope. Pools of two or three insects were placed into 2 mL microcentrifuge tubes, each one containing a 3 mm stainless steel bead, and homogenized at 30 Hz for 1 min 30 s by using a TissueLyser II system (Qiagen, Hilden, Germany). Samples were then incubated with 500 μL of CTAB buffer (3% cethyl-trimethyl-ammonium bromide, 100 mM Tris-HCl pH 8, 20 mM EDTA, 1.4 M NaCl) at 65 °C for 20 min. After the addition of 500 μL of 24:1 chloroform–isoamyl alcohol and a centrifugation at 15,800× *g* for 10 min, 320 μL of the aqueous phase were transferred into a new 1.5 mL microcentrifuge tube. Nucleic acids were precipitated with 500 μL of isopropanol and collected by centrifugation at 15,800× *g* at 4 °C for 20 min. Pellets were washed with 500 μL of 70% ethanol and dried after a centrifugation at 15,800× *g* at 4 °C for 15 min. Nucleic acids were dissolved in 100–150 μL of 10 mM Tris-HCl pH 8 and stored at −20 °C. The same protocol was followed for the DNA extraction from *N. campestris* and the isolated larvae of *V. aucta*, although nucleic acids were extracted from single individuals and finally dissolved in 30 μL of 10 mM Tris-HCl pH 8.

#### 2.2.3. Diagnostic PCRs

The concentration and purity of the extracted DNA were assessed with a NanoDrop 2000 spectrophotometer (Thermo Fisher Scientific, Waltham, MA, USA). The samples were then diluted with autoclaved ultrapure water to the concentration of 40 ng/μL DNA. Three microliters of the dilutions were used as templates in the diagnostic real-time PCR amplifications. The reaction mix contained 5 μL of 2× SsoAdvanced Universal SYBR mix (Bio-Rad, Hercules, CA, USA), 1 μL of 10 μM primer forward and 1 μL of 10 μM primer reverse. Specific primers targeting the *ITS2* region of *V. aucta* and oligonucleotides targeting the *18S rRNA* gene of the spittlebugs (Table 2) were used in parallel reactions to exclude the presence of PCR inhibitors and therefore a false negative diagnosis. Real-time PCR program consisted of 10 min at 95 °C, followed by 40 cycles in a series of 15 s at 95 °C, 1 min at 58 °C and plate read. At the end of the cycling program, the temperature was lowered to 60 °C and melting curves were obtained by a plate read, every increment of 0.3 °C. The positive control was represented by the diluted DNA extracted from a dissected *V. aucta* larva, devoid of host DNA.

To exclude false positive results, the samples that were detected over the 33rd cycle were tested with specific primers targeting the mitochondrial *COI* gene of *V. aucta* (Table 2) in conventional PCR. Two microliters of the diluted DNA (40 ng/μL) were added to the reaction mix containing 0.2 μL of 5 U/μL Taq polymerase and 5 μL of 10× buffer (Polymed, Italy), 2 μL of 50 mM MgCl_2_, 5 μL of 2 mM dNTPs, 5 μL of 10 μM primer forward, 5 μL of 10 μM primer reverse, and 25.8 μL of autoclaved ultrapure water. PCR program consisted of 5 min at 95 °C, followed by 35 cycles in a series of 30 s at 95 °C, 30 s at 58 °C, and 1 min at 72 °C (ending with 10 min at 72 °C).

To test the sensitivity of the assay, both pairs of primers targeting the *ITS2* region and the *COI* gene were tested in real-time and conventional PCR on a dilution of 0.05% *V. aucta* DNA in *P. spumarius* DNA, which corresponded to 36 pg of *V. aucta* DNA in the total reaction volume.

### 2.3. Rearing of Verrallia aucta and Philaenus spumarius

Seven *P. spumarius* (two males and five females) with swollen abdomens, longer than forewings (a sign of putative parasitization [24]), were collected in mid-September 2018 at Paderna (Table 1, Site code: F). These individuals were placed in a cage containing a layer of soil at the bottom and transplanted potted plants (*Avena sativa* L., *Lactuca sativa* L., *Taraxacum officinale* Weber ex Wiggers and *Trifolium pratense* L.) to allow the survival of spittlebugs. The rearing cage was placed outdoors in the shadow, in a garden of the University campus at Grugliasco (Site code: C). It was checked every 3–4 days in order to record the pupation time of *V. aucta* in the soil and maintained until the following year to record the appearance of the pipunculid flies.

### 2.4. Data Analysis

The number of positive insects—i.e., insects parasitized by *V. aucta*—in the PCR-tested multiple samples was estimated for each site and time period. The probability (*P*) of a single insect in a sample of being parasitized was estimated using the Swallow’s formula [31]:*P* = 1 − (1 − *H*)^1/*k*^(1)
where *H* is the proportion of negative samples, and *k* is the number of insects in each sample. The parasitization prevalence in each site and time period was obtained by multiplying the number of sampled insects for *P*.

Data analysis and maps were done using the packages tidyr and ggplot2 in the software R [32].

## 3. Results

### 3.1. Molecular Detection of Verrallia aucta

#### 3.1.1. Primer Specificity

The designed primers targeting the *ITS2* region and the *COI* gene of *V. aucta* amplified the DNA fragments of 183 and 308 bp, respectively (Table 2). Conventional PCR gave positive results when performed on DNA extracted from the larvae of *V. aucta*, whereas no amplicons were obtained when DNA extracted from non-parasitized spittlebugs was used as template. The *ITS2* and *COI* sequences obtained from purified PCR products perfectly matched with those found in GenBank and used for primer design (100% nucleotide identity). No differences between the nucleotide sequences of the pipunculid larvae isolated from *P. spumarius* and *N. campestris* were observed, indicating that *V. aucta* likely parasitizes both spittlebug species.

#### 3.1.2. Real-Time PCRs

Given their amplicon size, the primers targeting the *ITS2* region were best suited for use in real-time PCRs. The positive control was detected on average at the 15th amplification cycle, while field-collected parasitized samples were detected between the 12th and the 33rd cycle (Figure 1a), with a specific melting peak at 78.3 °C (Figure 1b). The *18S rRNA* gene of *P. spumarius* and *N. campestris* was detected between the 14th and the 16th amplification cycle (Figure 1c) with a specific melting peak at 84.9 °C (Figure 1d). When the sensitivity of the diagnostic assay was tested, the *V. aucta* DNA obtained from a single larva, diluted to a ratio of 1: 2000 in *P. spumarius* DNA, was detected at the 26th amplification cycle (*ITS2* amplicon) or visualized as a clear band on gel electrophoresis (*COI* amplicon).

### 3.2. Prevalence of the Parasitoid in the Field

#### 3.2.1. *Philaenus spumarius*

Spittlebug adults parasitized by *V. aucta* were identified in northwestern Italy from mid-June to the end of September, although a single parasitized female was collected in mid-October in Asti (Piemonte region; Site code: D). No parasitoids were detected in adults collected in late May, beginning of June, October, November, and December (Table 3). Therefore, the samples were collected only from July to September in northeastern Italy in 2018 and 2019, to estimate prevalence of *V. aucta* in this area (Table 4).

In northern Italy, the parasitoid was present both in plain–foothill and montane sites, above 1000 m a.s.l. A large variability in the parasitization prevalence was observed among the sites but the percentage of parasitized adults rarely exceeded 15% and when data were pooled over the different years, the highest parasitization rate (17.5%) was recorded in Paderna (Piemonte region; Site code: F; Figure 2). According to our surveys, in northeastern Italy (Veneto and Trentino-Alto Adige regions) the parasitoid was present but the parasitization rate of *P. spumarius* over the season did not exceed 5.6% (Figure 3).

Overall, no significant difference in parasitization prevalence between sexes was found (Chi-square = 1.58, df = 1, *p* = 0.208).

#### 3.2.2. *Neophilaenus campestris*

Adults of *N. campestris* were analyzed for the presence of *V. aucta* by dissection (samples collected from May to September) and PCR (samples collected in September). Only two adults were found parasitized, a female collected in late July at Prali (Table 1, Site code: G) and a male collected in September at Cocconato (Site code: D). The other analyzed adults of *N. campestris*, all collected in the Piemonte region (Site codes: C, *n* = 16; D, *n* = 13; F, *n* = 31; G, *n* = 1), were not parasitized by the pipunculid.

### 3.3. Biology of Verrallia aucta

#### 3.3.1. Preimaginal Stages

The egg of *V. aucta* could be dissected from a freshly parasitized female; it was 0.61 mm in length and 0.21 mm in width, elongated and oval in shape, rounded at one end and more tapered at the other one (Figure 4a). Young (first instar) larva has a terminal vesicle and lacks a posterior spiracular plate (Figure 4b,c). Dissected first instar larvae ranged from 1.3 to 2.78 mm (mean = 2.10, sd = 0.51, *n* = 9) in lenght and from 0.6 to 1.5 mm (mean = 1.07, sd = 0.33, *n* = 9) in width. Mature (second instar) larva has a posterior spiracular plate (Figure 4d). Dissected ones ranged from 2.75 to 3.89 mm (mean = 3.23, sd = 0.50, *n* = 7) in length, and from 1.74 to 2.38 mm (mean = 2.06, sd = 0.19, *n* = 7) in width. Terminal vesicle and posterior spiracular plate were discriminant features proposed by Whittaker [24] and Coe [33]. At the second larval stage, the host female spittlebugs showed degenerated ovarioles (Figure 4d).

#### 3.3.2. Life Cycle

Adults of *P. spumarius* were recorded in this study from the beginning of May to mid-December in the plain–foothill zone and from the beginning of July to mid-September in the montane zone. In the Piemonte region, an egg of *V. aucta* was observed in a female collected in the montane zone at the end of July. The larvae of *V. aucta* were found by dissecting adult bodies from the end of July until mid-October in the plain–foothill zone and from the beginning of August until mid-September in the montane zone.

Seven adults of *P. spumarius* with swollen abdomens were confined in rearing cages in mid-September 2018. On September 25, three mature larvae of *V. aucta* emerged from the dead hosts (one male and two females) and pupated in the soil at the bottom of the cage. On October 10th, one more larva of *V. aucta* emerged from a spittlebug female and pupated. The three remaining adults survived up to November 30 and no additional parasitoid pupae emerged. In spring 2019, a single adult of *V. aucta* emerged, on May 23. In the same rearing, adults of *P. spumarius* emerged between May 31st and June 3rd. In all four instances observed, the pipunculid larvae emerged from their hosts through the dorsal intersegmental sclerites between tergites four and five (Figure 4e).

Based on our observations, the detection of larvae inside the insect bodies (by both dissection and PCR) and literature data, we suggested a general scheme for the life cycle of *V. aucta* on *P. spumarius* in the plain–foothill zone of northern Italy (Figure 5).

## 4. Discussion

The presence of *V. aucta,* a parasitoid of adult *P. spumarius*, has been confirmed for Italy. Its occurrence was previously reported in the Fauna Europaea Online [26] and in the database of pipunculid species in Europe [34]. Apart from the aforementioned reports, no information is available on this parasitoid fly in Italy. Following the discovery of *X. fastidiosa* in Europe, namely of the subsp. *pauca* transmitted to olive by *P. spumarius*, interest in biological control agents of this pest has greatly increased and has prompted research on natural enemies, namely an egg parasitoid, *O. vulgatus* [20] and a generalist predator, *Z. renardii* [21]. *Verrallia aucta* does not rapidly kill the host, which can survive until late in the season. However, its effect on population dynamics/abundance over time, although delayed by one year, is equivalent to a quick-acting mortality factor, because parasitized females are sterile [24] [this work]. Parasitized individuals may survive for the whole season and possibly transmit the bacterium, but the following year, the vector population level will be lower because of parasitism.

We developed a new molecular tool for the specific identification of *V. aucta* in its spittlebug hosts, based on *ITS2* and *COI* sequences, previously used for the integrative taxonomy of *Verrallia* spp. [27]. The small size of the *ITS2* amplicon makes these primers perfectly suitable for a real-time PCR assay which is fast, sensitive and almost free of contaminations. The PCR assay was applied to a high number of field-collected samples and proved to be effective and specific, as no amplification signals were detected from non-parasitized insects. Compared to the dissection method—the conventional way to identify parasitized spittlebug adults—PCR is more sensitive, as it is able to also detect parasitoid eggs and early first instar larvae, and allows the simultaneous analysis of several individuals, because a single parasitized sample from a group analysis can be easily identified (*V. aucta* DNA from a mature larva can be diluted thousands of times in the spittlebug DNA and still be detected).

We provided evidence that *V. aucta* is not species-specific, since it was detected in both *P. spumarius* and *N. campestris*. Based on genomic data, our study does not agree with the study carried out in England by Whittaker [24], where *V. aucta* parasitizes *P. spumarius* and *N. lineatus*, whereas *N. campestris* is likely to be parasitized by *Verrallia setosa* Verrall. Indeed, our results indicate that the larva parasitizing *N. campestris* shares the identical *ITS2* and *COI* sequences of *V. aucta.* This is possibly due to different host associations of *V. aucta* throughout its range. However, based on the results of our study, which included a low number of specimens, we cannot draw general conclusions on the prevalence of *V. aucta* in *N. campestris* populations of northern Italy.

Our observations, surveys, and small-scale rearing, together with the very few data from the literature [24,35], suggest that *V. aucta* is monovoltine and synchronous with its host. The adults emerge at the same time as *P. spumarius* and *N. campestris* (i.e., starting from mid-May in northern Italy). The pipunculid then parasitizes newly emerged spittlebug adults, its larvae develop inside the body of the host throughout the summer and pupates in the soil at the beginning of autumn. Therefore, in northern Italy, the parasitoid survives in the soil as pupa from October to May.

The parasitism rates of *P. spumarius* in the different sites of northern Italy were quite low, ranging from 3.1% to 17.5% in northwestern regions and not exceeding 5–6% in the sites sampled in northeastern regions. According to Whittaker [24], the parasitization rate was constantly higher during a four-year period at Wytham Woods, Berkshire, UK, ranging from 30% to 40%. However, these data reflected the situation of one site only, and the same author noticed that in some sites of central and northern UK, Finland, and Holland, the pipunculid was not recorded, concluding that it was likely that the two host species of spittlebugs (*P. spumarius* and *N. lineatus*) were living in some areas outside the range of the parasitoid. No significant differences were recorded in the parasitism of the males and females of *P. spumarius*, contrary to the observations of Whittaker [24] that reported higher parasitization rates for males (46%) than females (31%).

The parasitization rate may be influenced by many factors, among which the level of disturbance of the environment, which is higher in a cropping system compared to a natural or semi-natural habitat. In particular, soil tilling during the October–May period can heavily affect the survival of pipunculid pupae, and insecticide treatments during the adult life of the fly may also contribute to suppress the parasitoid population. While the site where Whittaker recorded high rates of parasitization is a natural site [24], we collected most of the spittlebug samples in olive groves and vineyards. These agroecosystems, although organic, are more managed—e.g., the soil is periodically tilled—than the grassland of Berkshire. Still, the *V. aucta* prevalence recorded in western and eastern Alps in alpine grasslands, almost natural environments, was not higher than in the olive plantations and vineyards of the plain–foothill zones. In this respect, the effect of altitude on the distribution and abundance of the pipunculid fly cannot be excluded, as in those areas, due to prolonged snow coverage and low temperatures, the lifespan of adults *P. spumarius* is shorter (about three months, compared to 5–6 months in the plane), therefore the parasitoid should be able to complete its development in a shorter period and at a lower temperature. However, it is very likely that the *V. aucta* development is relatively independent from a thermal requirement (in degree days), as it is present in a wide geographical area, characterized by very different temperature conditions and where, apparently, it is always monovoltine and synchronous with its host [24,35]. The presence of *V. aucta* in Alpine sites at high elevation is consistent with its reported presence in northern Europe [34] and confirms that low temperatures are not a critical issue and its distribution is mainly limited by the presence of the hosts.

The pest status of *P. spumarius* in North America, caused by the high densities historically reached by this exotic spittlebug species [4,36], could be partly due to the absence of natural enemies like *V. aucta.* Indeed, as speculated by Whittaker [24], *P. spumarius* was probably introduced to the New World as overwintering eggs on plant material, losing its pipunculid parasitoid, which pupates in the ground in winter. The absence of this parasitoid, and possibly of other natural enemies, could have created an enemy release effect of the spittlebugs that translated into an increased demographic success [37]. Moreover, density-dependent mortality factors in the adult stage, strengthened by the presence of the pipunculid parasitoid *V. aucta* [25], may prevent high population fluctuations, whereas, in the absence of these, populations may be highly unstable, thus producing outbreaks as reported in North America [38,39].

Taking advantage of the fast and reliable diagnostic assay presented in this work, surveys should be carried out in new areas, including those where *X. fastidiosa* is epidemic, like in the Apulia region of Italy and other European foci, to achieve a broader picture of *V. aucta* presence and prevalence. Moreover, the sampling of adults should be planned to inspect different spatial levels (environments/landscapes/ecosystems), in order to gain insights on the environmental factors favoring or disadvantaging the parasitoid. Although in the investigated areas, parasitism rates were quite low, it is possible that, under proper environmental conditions, this parasitoid might be more efficient (as reported at Wytham Woods, Berkshire, UK [24]). Once the factors favoring the multiplication of this pipunculid under natural conditions are known, an inoculative biological control approach could be envisaged in areas where *V. aucta* is absent or poorly represented, by transferring parasitized adults into new sites and preserving suitable environmental conditions for its establishment and multiplication. The same should be done with the oophagous parasitoid *O. vulgatus*, which was found to be common and abundant in the Corsica island of France [20].

In order to preserve *V. aucta* in the areas where *X. fastidiosa* is spreading and the control of insect vectors is compulsory, control methods should target spittlebug nymphs in olive or almond groves, possibly limiting the soil tillage and preferring the mowing of weeds. Tilling the soil is more effective against *P. spumarius* nymphs compared to the mowing of weeds, but when performed in April it is likely to kill *V. aucta* pupae in the soil, compromising the natural control of the spittlebug populations. Moreover, insecticide applications against newly emerged adults of *P. spumarius* at the end of spring–beginning of summer, although rational in view of preventing *X. fastidiosa* transmission, might severely impact adults of *V. aucta* that are present in this period of the year [38]. A trade-off between these different control strategies should be reached, especially in areas with a high natural prevalence of *V. aucta,* to achieve the most effective control of the spittlebug. Moreover, *P. spumarius* being very polyphagous and widely present in uncultivated areas [8], refuges for the parasitoid should be available wherever crops are interspersed with non-cultivated areas. The ultimate goal of establishing and/or preserving cropping and non-cropping habitats suitable for the whole set of *P. spumarius* natural enemies should be part of a strategy aimed at preventing the spread of *X. fastidiosa*.

## 5. Conclusions

*Verrallia aucta*, a pipunculid parasitoid of the spittlebug vectors of *X. fastidiosa* in Europe, is present in several sites of northern Italy, both in plain–foothill and montane zones, although showing low prevalence. Since no evident sign of parasitism can be observed on insect hosts, we developed a tool to identify the parasitized spittlebug adults. Observations of the *V. aucta* life cycle, which is monovoltine and synchronous with its host, allowed us to propose a general scheme representing the *V. aucta* life cycle on *P. spumarius*.

## Figures and Tables

**Figure 1 insects-11-00607-f001:**
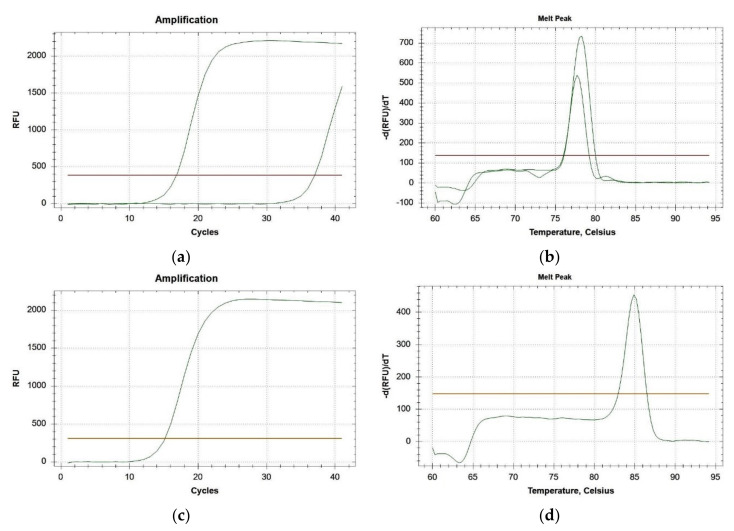
(**a**) Examples of amplification curves of *Verrallia aucta ITS2* region detected when DNA extracted from two parasitized samples was used as template in real-time PCR; (**b**) melting curves of the *ITS2* amplicons obtained from the same samples; (**c**) example of amplification curve of *Philaenus spumarius*/*Neophilaenus campestris 18SrRNA* gene amplified in real-time PCR; and (**d**) the melting curve of the *18SrRNA* amplicon.

**Figure 2 insects-11-00607-f002:**
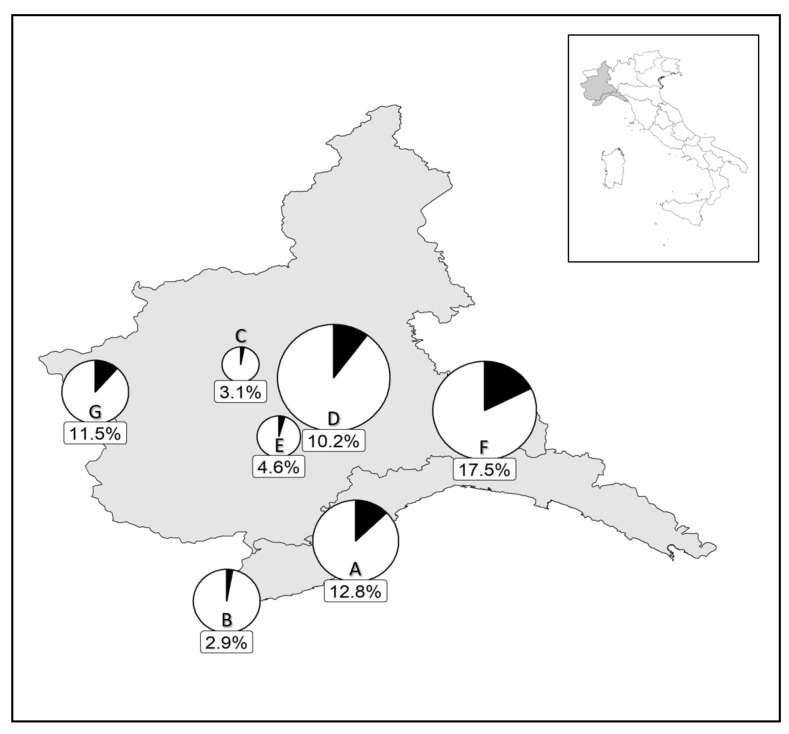
Percentage of *Philaenus spumarius* parasitized by *Verrallia aucta* in samples collected in northwestern Italian regions (Piemonte and Liguria) from June to October. Parasitization rates were calculated by pooling data from the dissections and molecular analyses of samples collected from 2016 to 2019. Pie chart size is a function of sample size: Arnasco-Finale Ligure (**A**), *n* = 172; Ventimiglia (**B**), *n* = 104; Chieri-Druento-Grugliasco (**C**), *n* = 32; Asti-Cocconato (**D**), *n* = 295; Cisterna d’Asti-Monteu Roero-Vezza d’Alba (**E**), *n* = 44; Paderna (**F**), *n* = 251; Cesana-Prali-Sestriere (**G**), *n* = 104.

**Figure 3 insects-11-00607-f003:**
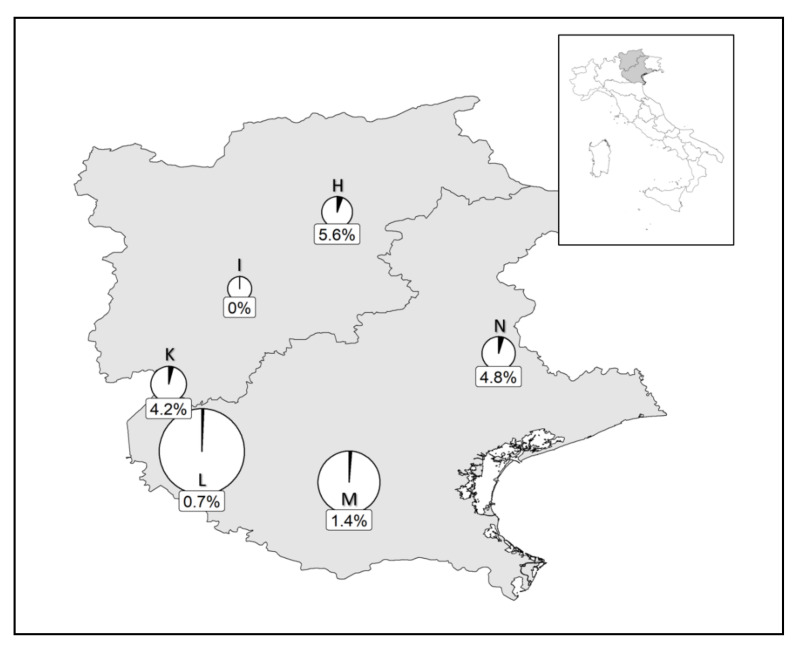
Percentage of *Philaenus spumarius* parasitized by *Verrallia aucta* in samples collected in northeastern Italian regions (Trentino-Alto Adige and Veneto) from July to September. Parasitization rates were calculated by pooling data from the dissections and molecular analyses of samples collected in 2018 and 2019. Pie chart size is a function of sample size: Castelrotto-Castelrotto San Michele-Compaccio (**H**), *n* = 18; Mezzocorona (**I**), *n* = 11; Malcesine (**K**), *n* = 24; Bussolengo-Montorio-Mezzane di Sotto (**L**), *n* = 138; Nanto-Arquà Petrarca-Lozzo Atesino (**M**), *n* = 73; Castello Roganzuolo (**N**), *n* = 21.

**Figure 4 insects-11-00607-f004:**
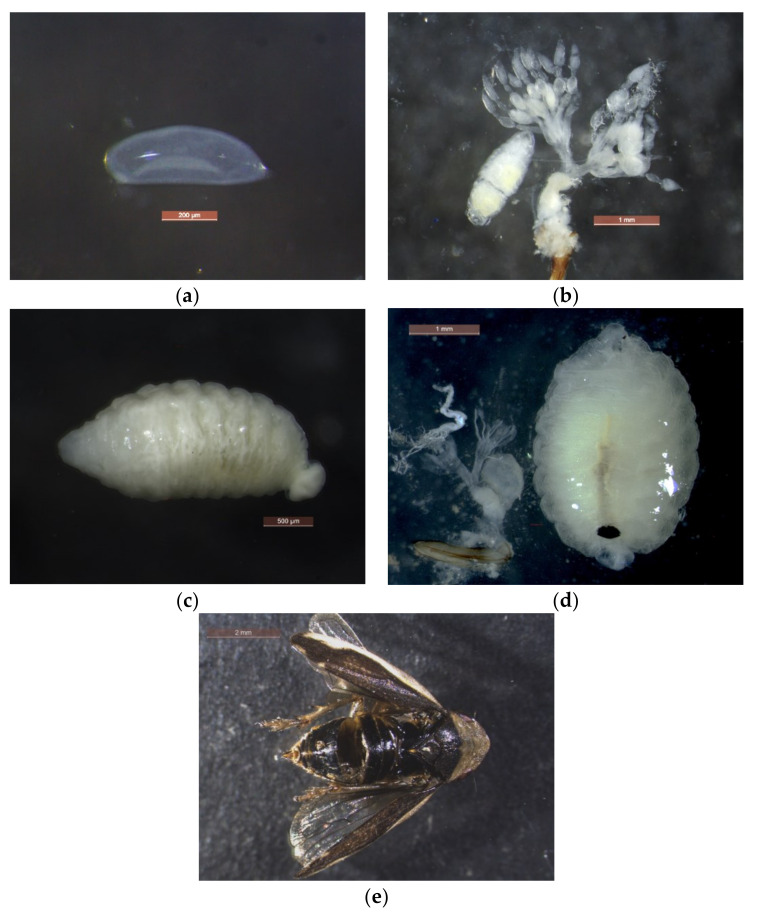
Developmental stages of the parasitoid *Verrallia aucta* in the host *Philaenus spumarius*: (**a**) egg dissected from a parasitized spittlebug female; (**b**) first instar larva (on the left) and ovarioles of the parasitized spittlebug female (on the right); (**c**) young larva with terminal vesicle on the right; (**d**) second instar larva with dark posterior spiracular plate (on the right) and degenerated ovarioles of the parasitized spittlebug female (on the left); and (**e**) the body of *P. spumarius* female after the emergence of *V. aucta* mature larva.

**Figure 5 insects-11-00607-f005:**
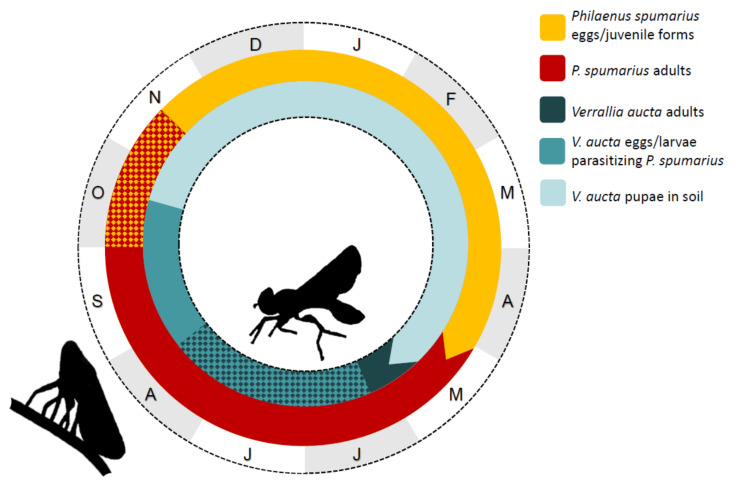
Life cycle of *Verrallia aucta* on *Philaenus spumarius* in the plain–foothill zone of northern Italy over the year (initials of months in capital letters).

**Table 1 insects-11-00607-t001:** Sites sampled for the presence of *Verrallia aucta* parasitizing adults of *Philaenus spumarius* and *Neophilaenus campestris*.

Locality	Region	Latitude	Longitude	Site Code	Altitudinal Zonation
Arnasco	Liguria	44.0766	8.1173	A	plain–foothill
Finale Ligure	Liguria	44.1811	8.3634	A	plain–foothill
Ventimiglia	Liguria	43.8073	7.5858	B	plain–foothill
Chieri	Piemonte	45.0154	7.7931	C	plain–foothill
Druento	Piemonte	45.1290	7.5880	C	plain–foothill
Grugliasco	Piemonte	45.0730	7.5874	C	plain–foothill
Asti	Piemonte	44.9194	8.1982	D	plain–foothill
Cocconato	Piemonte	45.0826	8.0594	D	plain–foothill
Cisterna d’Asti	Piemonte	44.8241	8.0117	E	plain–foothill
Monteu Roero	Piemonte	44.7784	7.9402	E	plain–foothill
Vezza d’Alba	Piemonte	44.7586	8.0220	E	plain–foothill
Paderna	Piemonte	44.8261	8.8947	F	plain–foothill
Prali	Piemonte	44.8759	7.0575	G	montane
Cesana	Piemonte	44.9530	6.7936	G	montane
Sestriere	Piemonte	44.9472	6.9054	G	montane
Castelrotto	Trentino-Alto Adige	46.5672	11.5560	H	montane
Castelrotto San Michele	Trentino-Alto Adige	46.5787	11.6021	H	montane
Compaccio	Trentino-Alto Adige	46.5410	11.6170	H	montane
Mezzocorona	Trentino-Alto Adige	46.2132	11.1455	I	plain–foothill
Malcesine	Veneto	45.7777	10.8206	K	plain–foothill
Bussolengo	Veneto	45.4475	10.8628	L	plain–foothill
Montorio	Veneto	45.4741	11.0768	L	plain–foothill
Mezzane di Sotto	Veneto	45.4837	11.1187	L	plain–foothill
Nanto	Veneto	45.4350	11.5791	M	plain–foothill
Arquà Petrarca	Veneto	45.2701	11.7395	M	plain–foothill
Lozzo Atestino	Veneto	45.2874	11.6195	M	plain–foothill
Castello Roganzuolo	Veneto	45.9182	12.3310	N	plain–foothill

**Table 2 insects-11-00607-t002:** Primers used in this study.

Primer	Sequence (5′–3′)	Target	Amplicon Size (bp)	Reference
VaITS2 Fw	TGCTGCTTGGACTACATATGG	*ITS2*	183	This work
VaITS2 Rv	AACGCATGGCACTAAACGAA			
VaCOI Fw	TGGAGGATTCGGAAACTGAC	*COI*	308	This work
VaCOI Rv	AGGTGATTCCTGTAGACCGC			
Mq Fw	AACGGCTACCACATCCAAGG	*18S*	98	[30]
Mq Rv	GCCTCGGATGAGTCCCG			

**Table 3 insects-11-00607-t003:** Monthly prevalence (%) of *Verrallia aucta* in the samples of *Philaenus spumarius* collected in northwestern Italian regions from 2016 to 2019 (sample size in brackets).

Region	Site	Year	MAY	JUN	JUL	AUG	SEP	OCT	NOV	DEC	Altitudinal Zonation
**Liguria**	**A**	**2016**	-	-	-	-	**16.7% ^d^** (6)	**0% ^d^** (8)	**0% ^d^** (6)	**0% ^d^** (3)	**plain–foothill**
		**2017**	0% ^d^ (5)	**0% ^d^** (13)	**0% ^d^** (10)	**16.7% ^d^** (12)	**10% ^d^** (10)	**0% ^d^** (11)	**0% ^d^** (10)	**0% ^d^** (7)	
		**2019**	-	-	**17.6% ^p^** (102)	-	-	-	-	-	
	**B**	**2018**	-	-	-	**2.9% ^p^** (104)	-	-	-	-	
**Piemonte**	**C**	**2016**	-	-	-	-	**0% ^d^** (1)	-	-	-	
		**2017**	-	**0% ^d^** (6)	**0% ^d^** (1)	-	-	-	-	-	
		**2018**	-	**0% ^d^** (5)	-	-	-	-	-	-	
		**2019**	0% ^d^ (5)	**33.3% ^d^** (3)	**0% ^d^** (7)	**0% ^d^** (2)	**0% ^dp^** (7)	-	-	-	
	**D**	**2016**	-	-	-	**8.7% ^d^** (23)	**0% ^d^** (15)	**7.7% ^d^** (13)	0% ^d^ (33)	0% ^d^ (3)	
		**2017**	0% ^d^ (30)	**0% ^d^** (11)	**15.4% ^d^** (13)	**14.3% ^d^** (7)	**0% ^d^** (11)	**0% ^d^** (16)	0% ^d^ (12)	-	
		**2018**	-	**0% ^d^** (3)	-	-	**15.6% ^dp^** (77)	-	-	-	
		**2019**	-	**15.9%**^dp^ (69)	**0% ^p^** (2)	-	**2.9% ^dp^** (35)	-	-	-	
	**E**	**2018**	-	-	-	-	**0% ^p^** (27)	-	-	-	
		**2019**	-	**40% ^dp^** (5)	**0% ^p^** (4)	-	**0% ^dp^** (8)	-	-	-	
	**F**	**2018**	-	**0% ^d^** (3)	-	-	**19.9% ^dp^** (136)	-	-	-	
		**2019**	-	**14.1% ^dp^** (64)	-	-	**14.6% ^dp^** (48)	-	-	-	
	**G**	**2016**	-	-	**0% ^d^** (9)	**0% ^d^** (6)	**0% ^d^** (2)	-	-	-	**montane**
		**2017**	-	-	**7.1% ^d^** (14)	**25% ^d^** (12)	**16.7% ^d^** (12)	-	-	-	
		**2018**	-	**0% ^d^** (1)	**4.5% ^d^** (22)	**14.3% ^d^** (14)	**0% ^d^** (2)	-	-	-	
		**2019**	-	-	**0% ^d^** (3)	**42.9% ^d^** (7)	-	-	-	-	

**^d^** Parasitization rate estimated by dissection, **^p^** Parasitization rate estimated by molecular analyses, **^dp^** Parasitization rate estimated by pooling data from dissections and molecular analyses.

**Table 4 insects-11-00607-t004:** Monthly prevalence (%) of *Verrallia aucta* in samples of *Philaenus spumarius* collected in northeastern Italian regions during summer 2018 and 2019 (sample size in brackets).

Region	Year	Site	JUL	AUG	SEP	Altitudinal Zonation
**Trentino-Alto Adige**	**2018**	**H**	**0%****^d^** (5)	**7.7%****^d^** (13)	-	**montane**
		**I**	**0%****^d^** (11)	-	-	**plain–foothill**
**Veneto**	**2019**	**K**	**0%****^p^** (3)	-	**4.8%****^p^** (21)	
		**L**	**3.7%****^p^** (27)	**0%****^p^** (43)	**0%****^p^** (68)	
		**M**	**0%****^p^** (5)	**0%****^p^** (32)	**2.8%****^p^** (36)	
		**N**	-	**8.3%****^p^** (12)	**0%****^p^** (9)	

**^d^** Parasitization rate estimated by dissection, **^p^** Parasitization rate estimated by molecular analyses.

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
