# Peer review of "Biology and Prevalence in Northern Italy of Verrallia aucta (Diptera, Pipunculidae), a Parasitoid of Philaenus spumarius (Hemiptera, Aphrophoridae), the Main Vector of Xylella fastidiosa in Europe"

_insects, 2020, doi:10.3390/insects11090607_

Round 1

Reviewer 1 Report

This is an interesting and well presented study. I can see no reason why it should not be accepted for publication.

Author Response

We thank the reviewer for her/his appreciation of the manuscript

Reviewer 2 Report

The manuscript provides information on the biology and prevalence of parasitoid, Verrallia aucta in Northern Italy. V. aucta is a parasitoid of hemipteran, Philaenus spumarius, which is the main vector of Xylella fastidiosa. The authors have focused on various aspects of biology and occurrence to gain more information on their population rate in Northern Italy. PCR and prevalence study results show that V. aucta is present in many sites including foothill and montane zones. The authors also developed a tool to identify parasitized spittlebug adults for future tests.

The paper is well written, provides detailed information with supporting figures. However, there are minor comments that need to be addressed. Please see below.

  • Why low numbers of campestris were used for dissection and molecular detection of V. aucta compared to the numbers of P. spumarius. The authors should have used at least 50 adults.
  • In section 2.3, where were the cages stored. In the green house? Provide more details including growing conditions (temperature, humidity, light if applicable).
  • In methods and results sections- the sensitivity of the assay can be combined with the previous section.
  • Lines 19, 32 – place ‘comma’ after spittlebug
  • Lines 55 – 58 – rephrase the sentence
  • Lines 62 – 66 – rephrase the sentence
  • Line 97 – I would suggest replacing singly by individually (optional)
  • Line 110 -a.s.l – expand when mentioning it for the first time
  • Line 115 – dissections
  • Line 214 – Figure 1 has very low resolution. I had a hard time to find the peaks in the graphs.
  • Line 248, 259 – The inset figure has low resolution. Either increase resolution or use colors for the lines and pie charts to stand out.

Author Response

We thank the reviewer for her/his appreciation and suggestions, that helped in improving the manuscript.

Why low numbers of campestris were used for dissection and molecular detection of V. aucta compared to the numbers of P. spumarius. The authors should have used at least 50 adults.

We agree in that the number of analyzed adults of N. campestris is very low compared to the one of Philaenus spumarius. We focused our work on this latter species and we collected N. campestris in order to clarify if Verrallia can parasitize this species, besides, P. spumarius. We believe this is the main result of N. campestris analyses. For this reason, we decided not to emphasize results of N. campestris parasitization and we summarized them in the text only, without a table or a figure.  However, it is worth noting that samplings of N. campestris took place only in the Piemonte region (to make clear this we have added a sentence in the Field Collection paragraph, line 112). When considering the sum of dissected insects (26) and the ones used for molecular detection (37), the sample size is well larger than 50, the minimum size in the reviewer opinion. Finally, we have added a sentence in the Discussion section (line 332) in order to better clarify the extent of our study on N. campestris.

In section 2.3, where were the cages stored. In the green house? Provide more details including growing conditions (temperature, humidity, light if applicable).

Details about the insect rearing conditions have been given in the Rearing paragraph (lines 178-179). Actually, the spittlebugs were kept outdoor, as the cages were stored in the shadow in the garden of the University.

In methods and results sections- the sensitivity of the assay can be combined with the previous section

As suggested, the paragraphs regarding the sensitivity of the molecular assay have been combined with the previous ones, both in the Materials and Methods (line 169) and in the Results (line 210) sections.

We have made some other modifications proposed by the reviewer, in particular:

  • a comma has been placed after “spittlebug” (lines 19 and 32);
  • the sentences starting at lines 55 and 62 have been rephrased;
  • we have replaced “singly” by “individually” (line 97);
  • by adding “above sea level”, we have explained “a.s.l.” when mentioned for the first time (line 110);
  • the plural form “Dissections” has been used in the title of the paragraph (line 115), as suggested.

Line 214 – Figure 1 has very low resolution. I had a hard time to find the peaks in the graphs.

In the pdf version of the file for review Fig. 1 has low resolution. However, the original files of the figure comply with the required resolution. However, we have improved the readability of Figure 1 (line 214), by darkening the lines on the graphs.

Line 248, 259 – The inset figure has low resolution. Either increase resolution or use colors for the lines and pie charts to stand out.

As for Figures 2 and 3 (line 249 and 259), we agree that in the pdf version of the manuscript, which we think was the one provided to the reviewers, their quality seemed poor. Actually, by uploading the image files, we had complied with the requirements of resolution of the journal (minimum 1,000 pixels width/height). Therefore, we have decided to maintain them as they were in the original submission.

Reviewer 3 Report

I found this to be an interesting paper and I enjoyed reading it.  I have a number of suggestions for the authors; these are almost entirely due to English not being their first language.  These are (somewhat) minor points but I believe they will enhance the readability of the manuscript.  Scientifically I find no fault with this manuscript.

Line 5:  Should order and family of the pathogen be included in the title?

Lines 30 & 31:  Disagreement of subject (larva) and verb (pupate).

Line 32:  Insert space between ")" and "(".

Line 34: "has been noticed since a long time' is a very awkward construction.  Maybe, "was first noticed a long time ago" would read better.

Also line 34: Insert space between ")" and "(".

Line 40: "In the four investigated years" is awkward.  Perhaps reword as, "During the four-year investigation".

Lines 45 & 46: Rewrite as "developing as an endoparasitoid"  and "overwintering as a pupa".

Line 53: There are more authors for this species than just Wells.

Line 119: Change "parasitoid" to "parasitoids"

Line 162: "33rd", not "33th"

Line 170: This might read better as "both pairs of primers"

Line 222: Insert "the" so this reads, "the end of September."

Lines 251 & 262: Change to "Pie chart size is a function of ...."

Line 268: I suggest replacing the comma (,) with a semicolon (;).

Lines 270 & 273: Delete "shows" and replace with a different word.  Maybe "has a".

Line 288:  I would write that no more or no additional parasitoid pupae emerged.

Line 290:  Disagreement between noun (larvae, plural) and pronoun (its, singular).

Line 305:  Rewrite - "a parasitoid of adult P. spumarius"

Line 309:  Delete "the" - it is unnecessary.

Line 322:  Insert "the" - "Compared to the dissection method ...."

Line 327: Delete the "s" - "evidence"

Line 329: Replace "while" with "whereas"

Line 333:  Insert a comma after "surveys"

Line 344: Insert a comma after "Finland"

Line 346: Insert "of" between "females" and "P."

Line 387:  Replace "will be" with "are"

Line 401:  Transpose "being" and "P. spumarius"

Lines 404-406:  The authors might consider joining this one-sentence paragraph to the preceding paragraph.

Author Response

We thank the reviewer for her/his appreciation and suggestions that helped to improve the quality of the manuscript.

Line 5:  Should order and family of the pathogen be included in the title?

Considering the journal guidelines and the international standards, we think that the order and family of the pathogen (X. fastidiosa) should not be mentioned, thus we have not made any modification to the title.

Lines 30 & 31:  Disagreement of subject (larva) and verb (pupate).

we have set the verb at the third person singular form (“pupates”) at line 31;

Line 32:  Insert space between ")" and "(".

space have been inserted

Line 34: "has been noticed since a long time' is a very awkward construction.  Maybe, "was first noticed a long time ago" would read better.

the sentence has been changed according to the reviewer suggestion

Also line 34: Insert space between ")" and "(".

space have been inserted

Line 40: "In the four investigated years" is awkward.  Perhaps reword as, "During the four-year investigation".

the sentence has been changed according to the reviewer suggestion

Lines 45 & 46: Rewrite as "developing as an endoparasitoid"  and "overwintering as a pupa".

the sentence at lines 45-46 has been corrected accordingly

Line 53: There are more authors for this species than just Wells.

Now it is “Wells et al.” in order to include all the authors of the species (line 33 and 53)

Line 119: Change "parasitoid" to "parasitoids"

OK, done

Line 162: "33rd", not "33th"

OK, done

Line 170: This might read better as "both pairs of primers"

“couples” has been replaced by “pairs”

Line 222: Insert "the" so this reads, "the end of September."

OK, done

Lines 251 & 262: Change to "Pie chart size is a function of ...."

figure captions have been corrected at lines 251 and 262

Line 268: I suggest replacing the comma (,) with a semicolon (;).

OK, done

Lines 270 & 273: Delete "shows" and replace with a different word.  Maybe "has a".

We changed as suggested

Line 288:  I would write that no more or no additional parasitoid pupae emerged.

“further” has been changed to “additional”

Line 290:  Disagreement between noun (larvae, plural) and pronoun (its, singular).

“Its” has been replaced by “their”

Line 305:  Rewrite - "a parasitoid of adult P. spumarius"

OK, done

Line 309:  Delete "the" - it is unnecessary.

OK, done

Line 322:  Insert "the" - "Compared to the dissection method ...."

OK, done

Line 327: Delete the "s" - "evidence"

OK, done

Line 329: Replace "while" with "whereas"

OK, done

Line 333:  Insert a comma after "surveys"

Line 344: Insert a comma after "Finland

we have inserted a comma after “surveys” (line 333)  and after “Finland” (line 344);

Line 346: Insert "of" between "females" and "P."

OK, done

Line 387:  Replace "will be" with "are"

OK, done

Line 401:  Transpose "being" and "P. spumarius"

 “being” has been moved after “P. spumarius

Lines 404-406:  The authors might consider joining this one-sentence paragraph to the preceding paragraph.

We have joined the last sentence to the preceding paragraph, as suggested by the reviewer.